# An integrative analysis of *5HTT*-mediated mechanism of hyperactivity to non-threatening voices

Chenyi Chen [1,2,3,4,5,10]✉, Róger M. Martínez [2,10], Tsai-Tsen Liao[6,10], Chin-Yau Chen[7], Chih-Yung Yang[8] & Yawei Cheng[1,8,9]✉

The tonic model delineating the serotonin transporter polymorphism's (*5-HTTLPR*) modulatory effect on anxiety points towards a universal underlying mechanism involving a hyper-or-elevated baseline level of arousal even to non-threatening stimuli. However, to our knowledge, this mechanism has never been observed in non-clinical cohorts exhibiting high anxiety. Moreover, empirical support regarding said association is mixed, potentially because of publication bias with a relatively small sample size. Hence, how the *5-HTTLPR* modulates neural correlates remains controversial. Here we show that *5-HTTLPR* short-allele carriers had significantly increased baseline ERPs and reduced fearful MMN, phenomena which can nevertheless be reversed by acute anxiolytic treatment. This provides evidence that the *5-HTT* affects the automatic processing of threatening and non-threatening voices, impacts broadly on social cognition, and conclusively asserts the heightened baseline arousal level as the universal underlying neural mechanism for anxiety-related susceptibilities, functioning as a spectrum-like distribution from high trait anxiety non-patients to anxiety patients.

[1] Department of Physical Medicine & Rehabilitation, National Yang-Ming University Hospital, Yilan, Taiwan. [2] Graduate Institute of Injury Prevention and Control, College of Public Health, Taipei Medical University, Taipei, Taiwan. [3] Research Center of Brain and Consciousness, Shuang Ho Hospital, Taipei Medical University, New Taipei City, Taiwan. [4] Graduate Institute of Mind, Brain and Consciousness, College of Humanities and Social Sciences, Taipei Medical University, Taipei, Taiwan. [5] Cell Physiology and Molecular Image Research Center, Wan Fang Hospital, Taipei Medical University, Taipei, Taiwan. [6] Graduate Institute of Medical Sciences, College of Medicine, Taipei Medical University, Taipei, Taiwan. [7] Department of Surgery, National Yang-Ming University Hospital, Yilan, Taiwan. [8] Department of Education and Research, Taipei City Hospital, Taipei, Taiwan. [9] Institute of Neuroscience and Brain Research Center, National Yang-Ming University, Taipei, Taiwan. [10] These authors contributed equally: Chenyi Chen, Róger M. Martínez, Tsai-Tsen Liao. ✉email: chenyic@tmu.edu.tw; ywcheng2@ym.edu.tw

In recent years, extant literature has pointed towards the possibility of a universal underlying mechanism involving a hyper-or-elevated baseline level of arousal and response even to neutral, non-threatening stimuli as the root of all anxiety-related ailments, those illnesses comorbid with anxiogenic symptomatology, or the susceptibility towards them (e.g., generalized anxiety disorder[1], post traumatic stress disorder[2,3] obsessive compulsive disorder[3,4], autism spectrum disorder[5], and schizophrenia[6,7]). Such potential mechanism has a basis in the 'tonic' model of *5-HTT*-dependent modulation of neural activity[8]. The *5-HTT* is a functional polymorphism in the serotonin-transporter-linked polymorphic region (5-HTTLPR) of the serotonin transporter gene (*SLC6A4*), and has been regarded as a potential genetic contributor for certain propensity towards anxiety-related traits since its discovery[9]. Specifically, since the *5-HTTLPR* short (S) – relative to long (L)– variant encodes less quantity of *5-HTT* mRNA and protein, consequently transporting less serotonin from the synaptic cleft back to the pre-synaptic neuron[8], it is of the common view that the *5-HTTLPR* short variant is related to mechanisms of negative emotionality, conferring some susceptibility towards certain affective disorders[10–12].

This assumption arises from the Differential Susceptibility Hypothesis[13], which explains how individual experience life circumstances and events in a differing manner, dependent on pre-existing biological factors which may result in certain predispositions. In our case punctually, from the alleged association between the short variant of the polymorphism and the observed increment in amygdala activation as a function of possessing such allele[14]. The aforementioned "tonic" model explains this *5-HTT*-dependent modulation of neural activity, and its consequences, by proposing that a high level of amygdala reactivity is present at baseline in individuals prone to anxiety-related personality traits, thus being more likely to process even non-threatening or neutral stimuli in a threatening manner[15,16]. This model is the counterpart to the "phasic" model, which explains the higher negative emotionality in S allele carriers as a function of higher-than-normal responses to threatening or aversive stimuli per se[8].

However, this mechanism has never been proven in non-clinical cohorts suffering from high anxiety symptomatology that not yet meet the criteria for diagnosis, as empirical support testing the interaction between the S allele and amygdala reactivity is mixed. Despite the first study observing heightened amygdala reactivity in carriers of the S allele and reporting that the *5-HTTLPR* accounted for more than 20% of the variance for the observed arousal[17], and although several studies subsequently replicated this association[15,18,19], others reported contradictory findings[14,20,21]. This partly due to the attentional modulation contributing to the variation of amygdala reactivity to threatening (angry and fearful) faces in a widely used emotional-face-matching paradigm[22,23], and partly due to the differing results in regards to the percentage of allelic variance accounting for the gene-amygdala association. While one meta-analysis reported that the polymorphic variance could account for approximately 10% of the association between the *5-HTTLPR* and heightened amygdala reactivity[24], another warned that the estimates might be distorted because of publication bias caused by relatively small sample sizes[14]. What's more, one study went as far as to cast doubt on the previously reported substantial effects, suggesting that the association of the *5-HTTLPR* variation with amygdala reactivity should be either much smaller, conditional, or even nonexistent[20]. Hence, how this polymorphism modulates neural correlates remains controversial. But given the impact of "epigenetic" mechanisms that encode environmental information from both internal and external bodily sources, the Gene–Brain interactions may render various degree of sensitivity towards threat processing[13].

Human voices, similar to faces, convey a wealth of social information[25]. Mismatch negativity (MMN), a component of the event-related potentials (ERPs), is elicited by a passive auditory oddball paradigm where participants engage in a task and ignore the stimuli that are presented to them in a random series, with one stimulus (standard) occurring more frequently than the others (deviant)[26]. MMN has been successfully utilized to establish a positive relationship between MMN amplitudes and the susceptibility towards anxiety symptomatology, as it is presumed to reflect the emotional hypervigilance characteristics of anxiety[27,28]. Findings which are more in tune with the phasic model of *5-HTT*-dependent neural modulation. This is due to MMN being able to index the biological mechanisms that sit in the border between automatic and attention-dependent processes, which control the gateways to conscious perception and higher orders of memory[29]. Consequently, because of this ability to tap on to attentional processes and memory, it has been argued that emotional MMN (eMMN)—which is a MMN subtype that makes use of emotionally spoken syllables embedded in the auditory oddball paradigm as the deviant stimuli triggering the MMN[30]—can assess the automatic neural processing of emotional voices in as early as the pre-attentive stage[31,32]. Furthermore, a processing chain that proceeds from the primary auditory pathway to brain structures implicated in cognition and emotion—e.g., the, orbitofrontal cortex, amygdala, superior temporal gyrus and sulcus—as well as in the saliency network (insula), has been revealed[31,33–35]. Accordingly, and interestingly enough, eMMN amplitudes become atypical and, in contrast to pure tone MMN, negatively associated to the concurrent symptomatology in youths with autistic traits[36], in adolescents with conduct disorder symptoms[37], and in patients with schizophrenia[38]. Particularly, MMN to threatening syllables significantly elicits amygdala activation[34]. In the case of fearful MMN, it has been observed as able to predict anxiety-related symptomatology[31]. Together, these findings provide support for the notion that eMMN can probe voice processing per se, disentangling emotional salience from attentional modulation. We thus suppose that eMMN can very well reflect *5-HTT*-dependent neural modulation.

The two major aims of this study are: (1) to test for a possible universal underlying mechanism generating negative emotionality and thus certain susceptibility towards anxiety; and (2) to replicate and extend the knowledge on *5-HTT*-dependent neural modulation in the debate of gene × environment interaction, more specifically the association between the *5-HTTLPR* and its imputed role in the propensity towards mental disorders[21]. By incorporating multimodal indices—including genetic, neurophysiological, biochemical, neuropharmacological, and behavioral measurements—, this study explores the possibility of an elevated level of arousal and response even to non-threatening stimuli already present at baseline as the universal mechanism behind anxiety-related ailments, or illnesses comorbid with anxiogenic symptomatology or the susceptibility towards them. Accordingly, this study genotyped the *5-HTTLPR*, and recorded the MMN in response to emotionally-spoken syllables in healthy volunteers who varied in trait and state anxiety. If the *5-HTTLPR* is able to genetically bias eMMN, we hypothesized that S allele carriers would exhibit distinct eMMN from noncarriers, and that eMMN would be associated with certain proclivity towards anxiety. Based on the two models of *5-HTT*-dependent neural modulation, we further hypothesized that if the phasic model is favored, then S allele carriers would show stronger eMMN than noncarriers. Alternatively, if the tonic model is held true, S allele carriers would show weaker eMMN as a function of an increase in ERPs to baseline neutral stimuli, finding which would also point emphatically towards the proposed heightened arousal at baseline level as the universal underlying neural mechanism incurring in

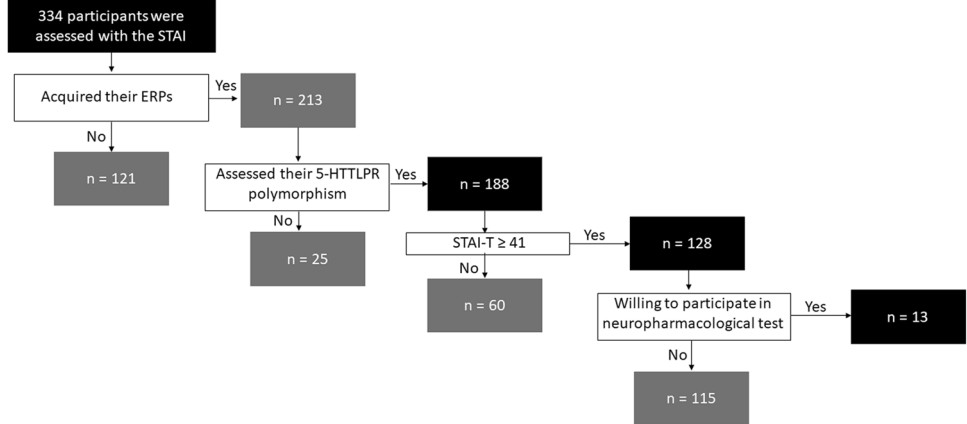

**Fig. 1 Flow-chart of participant selection for the study.** This study assessed state and trait anxiety (STAI) in three hundred thirty-four healthy volunteers, aged between 19–63 (mean ± SD: 27.1 ± 9.9, 155 males) years old. Subsequently, we genotyped the 5-HTTLPR and recorded the event-related potentials (ERPs) in one hundred eighty-eight of them, aged between 19–46 (23.4 ± 3.5, 92 males) years old, and which were included in the data analysis. One hundred and twenty-eight of these exhibited high trait anxiety scores (STAI-T ≥ 41), and which were used for the path analysis. Finally, 13 of them went ahead to participate in the neuropharmacological testing.

propensities towards anxiety, which would function as a spectrum-like distribution from anxiety patients to non-patients with high trait anxiety. Additionally, we conducted the follow-up neuropharmacological test on volunteers with high anxiety scores to see whether the hyper arousal and response to non-threatening stimuli could be reversed by acute anxiolytic treatment. While the phasic model predicts reduced eMMN resulting exclusively from decreased fearful deviant ERPs, the tonic model predicts reduced neutral standard ERPs.

## Results

**Genotyping distribution and behavioral performance**. This study assessed state and trait anxiety (STAI) in three hundred thirty-four healthy volunteers, aged between 19–63 (mean ± SD: 27.1 ± 9.9, 155 males) years old, as well as genotyped the 5-HTTLPR and recorded the eMMN in one hundred eighty-eight of them, aged between 19–46 (23.4 ± 3.5, 92 males) years old. One hundred and twenty-eight of which exhibited high trait anxiety scores (STAI-T ≥ 41)[39], but without reaching clinical significance (Fig. 1). The 5-HTTLPR was found to have allele frequencies of S, $n = 249$ (66.6%); LA, $n = 45$ (12%); and LG, $n = 80$ (21.4%), and a genotype distribution of S/S, $n = 88$ (46.5%); S/LG, $n = 47$ (25.1%); S/LA, $n = 28$ (15%); LG/LG, $n = 11$ (5.9%); LG/LA, $n = 11$ (5.9%); and LA/LA, $n = 3$ (1.6%). The genotype distribution of the 5-HTTLPR across all participants was in Hardy-Weinberg equilibrium, $\chi^2(3) = 1.99$, $P = .57$. The following analyses employed the genotype groups: S/S = 88, L/S = 75, and L/L = 25 (Table 1). The 5-HTTLPR genotype were not different across age ($P = .18$), gender (male % of total: 51.1% vs. 44.0% vs. 56.0%; $P = .40$), STAI-T ($P = .59$), and STAI-S ($P = .91$).

**Neurophysiological measures of pre-attentive discrimination**. MMN was determined by subtracting the neutral ERPs from angry and fearful ERPs (Fig. 2a). The four-way mixed ANOVA revealed main effects of deviant type (fearful vs. angry) ($F_{1, 179} = 10.35$, $P = 0.002$, $\eta p^2 = 0.055$, $(1−\beta) \approx 100\%$), coronal site (left, midline, right) ($F_{2, 358} = 15.91$, $P < 0.001$, $\eta p^2 = 0.082$, $(1−\beta) \approx 100\%$), gender (male vs. female) ($F_{1, 179} = 6.16$, $P = 0.014$, $\eta p^2 = 0.033$, $(1−\beta) \approx 100\%$), and genotype (L/L, L/S, S/S) ($F_{2, 179} = 8.28$, $P < 0.001$, $\eta p^2 = 0.085$, $(1−\beta) \approx 100\%$). Fearful MMN had significantly higher amplitudes than angry MMN. The S allele carriers exhibited weaker eMMN than did noncarriers, irrespective

### Table 1 Demographic and descriptive statistics of each genotype group for ERP analysis.

|  | L/L $n = 25$ 14 males | | L/S $n = 75$ 33 males | | S/S $n = 88$ 45 males | |
|---|---|---|---|---|---|---|
|  | **Mean** | **SD** | **Mean** | **SD** | **Mean** | **SD** |
| Age (years) | 22.5 | 2.6 | 23.9 | 3.7 | 23.2 | 3.6 |
| STAI-T | 42.7 | 8.5 | 44.9 | 9.5 | 44.6 | 9.2 |
| STAI-S | 37.4 | 7.2 | 36.9 | 7.9 | 37.4 | 9.2 |
| Fearful MMN (μV) | 5.03 | 3.56 | 3.27 | 2.46 | 3.07 | 2.47 |
| Angry MMN (μV) | 4.36 | 3.28 | 2.71 | 2.33 | 2.86 | 2.41 |

of the deviant type (Fig. 2b). Females (4.05 ± 0.27 μV) had significantly stronger MMN than did males (3.14 ± 0.25 μV).

Significant interactions were observed among the deviant type, coronal site, and genotype ($F_{4, 358} = 3.1$, $P = 0.017$, $\eta p^2 = 0.033$, $(1−\beta) \approx 100\%$), among the anterior-posterior site, coronal site, and genotype ($F_{4, 358} = 2.64$, $P = 0.035$, $\eta p^2 = 0.029$, $(1−\beta) \approx 100\%$), and among the coronal site, gender, and genotype ($F_{4, 358} = 3.58$, $P = 0.008$, $\eta p^2 = 0.038$, $(1−\beta) \approx 100\%$). Post hoc analyses revealed that the S/S exhibited a significant interaction between the deviant type and coronal site ($F_{2, 168} = 6.93$, $P = 0.002$, $\eta p^2 = 0.076$, $(1−\beta) \approx 100\%$); however, the L/L ($F_{2, 46} = 2.15$, $P = 0.15$) and L/S ($F_{2, 144} = 1.99$, $P = 0.15$) did not. At the right electrodes, fearful and angry MMNs were comparable in the S/S (2.98 ± 0.24 vs. 2.94 ± 0.23 μV; $t_{86} = 0.235$, $P = 0.81$); however, they differed in the L/L (5.4 ± 0.73 vs. 4.56 ± 0.71 μV; $t_{24} = 2.8$, $P = 0.01$) and L/S (3.13 ± 0.26 vs. 2.71 ± 0.25 μV; $t_{74} = 1.97$, $P = 0.053$), with larger amplitudes in fearful than angry MMN. Post hoc analyses revealed that the effect size of the interaction between gender and genotype varied along the factor of coronal site (left: $F_{2, 179} = 3.17$, $P = 0.045$, $\eta p^2 = 0.034$; midline: $F_{2, 179} = 6.20$, $P = 0.002$, $\eta p^2 = 0.065$; right: $F_{2, 179} = 2.86$, $P = 0.06$, $\eta p^2 = 0.031$). The gender effect was the strongest in the midline electrodes and exclusively found in participants with the L/L variant (females vs. males: 6.76 ± 0.69 vs. 3.20 ± 0.61 μV), but not in those with the L/S (3.02 ± 0.33 vs. 2.95 ± 0.38 μV) nor the S/S variant (2.92 ± 0.34 vs. 2.96 ± 0.32 μV) (Supplementary Fig. 1).

To examine whether eMMN was affected by neutral standards or fearful deviants, correlation analyses were conducted against ERPs and MMN amplitudes. Fearful MMN was positively

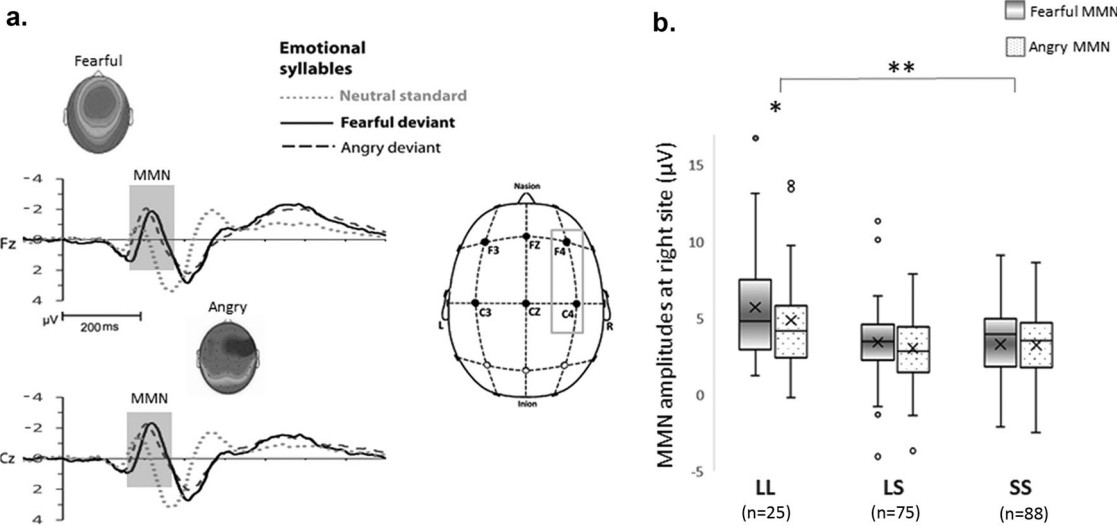

**Fig. 2 Association between the 5-HTTLPR and emotional MMN. a** Subtracting neutral ERPs from fearful and angry ERPs determines fearful and angry MMN, respectively. **b** L/L homozygotes (4.7 ± 0.44 μv) exhibit stronger MMN than S allele carriers (LS: 2.99 ± 0.25 μv; SS: 2.96 ± 0.24 μv), irrespective of the deviant type. There are significant interactions among the deviant type, coronal site, and genotype ($F_{4, 368} = 3.43$, $P = 0.01$, $\eta p^2 = 0.036$, $(1-\beta) \approx$ 100%). Post hoc analyses indicate that, at F4 and C4 electrodes, fearful and angry MMN are comparable in the S/S group (2.98 ± 0.24 vs. 2.94 ± 0.23 μV: $t_{86} = 0.24$, $P = 0.81$), but significantly different in the L/L (5.4 ± 0.73 vs. 4.56 ± 0.71: $t_{24} = 2.80$, $P = 0.01$) and L/S (3.13 ± 0.26 vs. 2.71 ± 0.25: $t_{74} = 1.97$, $P = 0.053$), with larger amplitudes in fearful than angry MMN.

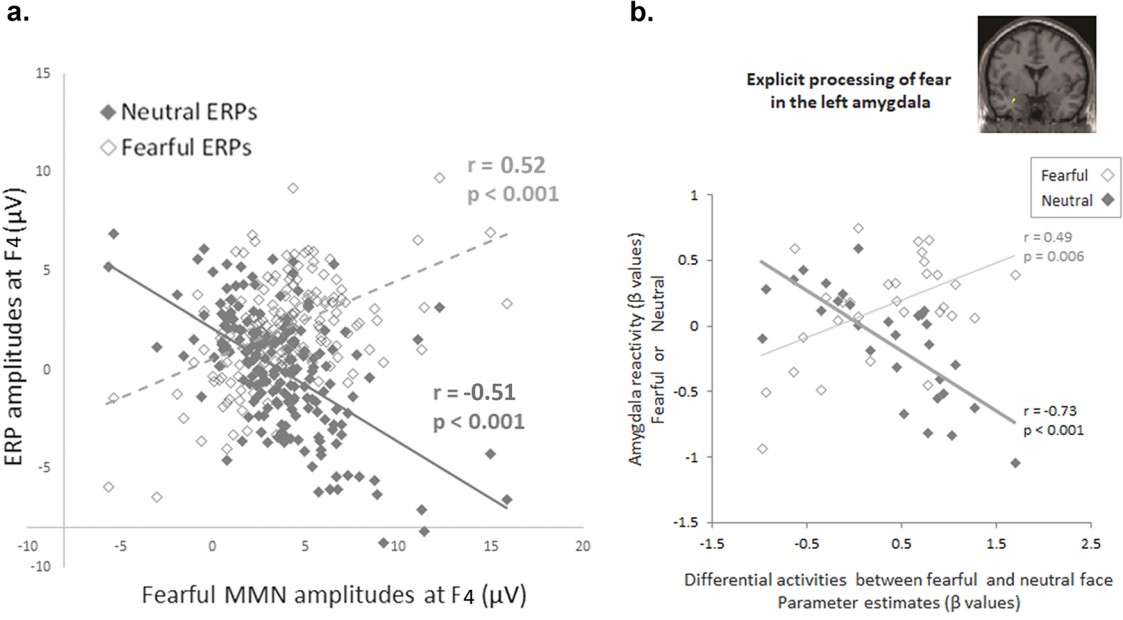

**Fig. 3 Fearful MMN as a function of neutral and fearful ERPs. a** Fearful MMN amplitudes are positively correlated with fearful ERP ($r_{187} = 0.47$, $P <$ 0.001), and negatively correlated with neutral ERP ($r_{187} = -0.57$, $P < 0.001$). Fisher r-to-z transformation confirmed that both of fearful and neutral ERPs independently contribute to fearful MMN ($\Delta z = 11.1$, $P < 0.01$). Larger fearful MMN are ascribed to increased fearful as well as to reduced neutral ERP amplitudes. **b** The same pattern emerges in an independent dataset ($n = 30$, 16 males). The negative emotionality (fearful vs. neutral) in the amygdala varies as a function of neutral and fearful face processing in a previously collected dataset[31]. The amygdala reactivity to explicitly perceived emotionality (fearful vs. neutral) is positively correlated with the response to fearful faces ($r = 0.49$, $P = 0.006$) but negatively correlated with the response to neutral faces ($r = -0.73$, $P < 0.001$). Fisher r-to-z transformation confirms that both of the responses to fearful and neutral faces independently contribute to explicitly perceived emotionality ($\Delta z = 5.38$, $P < 0.01$). Meanwhile, the amygdala reactivity to fearful facial expressions (explicit fear vs. neutral faces) is closely associated with MMN to fearful vocal expressions (fearful vs. neutral voices) ($r = 0.58$, $P = 0.008$).

correlated with the amplitudes of fearful ERP ($r_{187} = 0.52$, $P <$ 0.001) and negatively correlated with neutral ERP ($r_{187} = -0.51$, $P < 0.001$). Fisher r-to-z transformation confirmed that both of responses to fearful and neutral ERPs independently contributed to the fearful MMN ($\Delta z = 11.1$, $P < 0.01$). Larger fearful MMN was ascribed to increased fearful as well as reduced neutral ERP amplitudes (Fig. 3a). To bolster confidence in the anxiety-related hyper-responsiveness during the baseline condition, we attempted to replicate the findings in an independent dataset. Specifically, we identified a close relationship between eMMN to fearful vocal expressions (fearful vs. neutral voices) and left amygdala reactivity to fearful facial expressions (explicit fear vs.

neutral) ($r = 0.49$, $P = 0.006$ and $r = -0.73$, $P < 0.001$, respectively) (Fig. 3b) in a previously collected dataset ($n = 30$, 16 males)[31], where the amygdala activity to neutral faces varied along with trait anxiety. The same pattern emerged in an independent dataset. The negative emotionality (fearful vs. neutral) in the amygdala varied as a function of neutral and fearful face processing in a previously collected dataset. The amygdala reactivity to explicitly perceived emotionality (fearful vs. neutral) was positively correlated with the response to fearful faces ($r = 0.49$, $P = 0.006$) but negatively correlated with the response to neutral faces ($r = -0.73$, $P < 0.001$). Fisher $r$-to-$z$ transformation confirmed that both of the response to fearful and neutral faces independently contributed to explicitly perceived emotionality ($\Delta z = 5.38$, $P < 0.01$). Meanwhile, the amygdala reactivity to fearful facial expressions (explicit fear vs. neutral faces) was closely associated with MMN in response to fearful vocal expressions (fearful vs. neutral voices) ($r = 0.58$, $P = 0.008$). Path analyses showed that the lowest BIC value (BIC = 2.491), i.e., the optimal fit, was obtained for the model with paths from 5-HTTLPR to fearful MMN and from fearful MMN to STAI-S (5-HTTLPR → fearful MMN → STAI-S). 5-HTTLPR explained 5.02% of the variance in fearful MMN, and fearful MMN explained 1.44% of the variance in STAI-S when the variance shared between 5-HTTLPR and STAI-S was partialled out (Supplementary Fig. 2, Supplementary Table 1).

Accordingly, to test whether the reduced eMMN in the S allele carriers resulted from the altered ERP responses to neutral syllables among individuals with high trait anxiety, a one-way ANOVA comprising genotype (L/L, L/S, and S/S) as the between-subjects factor was performed at the right electrodes for participants with STAI-T ≥ 41 ($n = 128$). There was a significant main effect of genotype (L/L, L/S, and S/S) ($F_{2, 125} = 3.51$, $P = 0.033$, $\eta p^2 = 0.053$, $(1-\beta) \approx 66.5\%$). Post hoc analyses showed that the S/S ($0.55 \pm 0.36\,\mu V$) exhibited larger neutral ERPs than the L/S ($-0.33 \pm 0.39\,\mu V$) and L/L ($-1.52 \pm 0.76\,\mu V$). The S allele carriers exhibited weaker eMMN and stronger neutral ERPs as compared with noncarriers (Fig. 4). Additionally, to test whether this pattern was associated with trait anxiety, we examined the relationship of STAI-T with eMMN and neutral ERPs. In individuals with high trait anxiety, their correlation was not

significant (all $P > 0.1$). However, in those with low trait anxiety (STAI-T < 41), the STAI-T scores were negatively correlated with the eMMN amplitudes (angry: $r_{60} = -0.35$, $P = 0.003$; fearful: $r_{60} = -0.24$, $P = 0.031$), and positively correlated with the neutral ERPs amplitudes ($r_{60} = 0.25$, $P = 0.028$, one-tailed).

**Anxiolytic effect on neutral ERPs**. While acute lorazepam treatment had no significant effect in the BAI ($t_{12} = 1.35$, $P = 0.20$), STAI-S ($t_{12} = 1.79$, $P = 0.10$), STAI-T ($t_{12} = 1.2$, $P = 0.26$), fearful MMN ($F_{1, 12} = 1.89$, $P = 0.19$), angry MMN ($F_{1, 12} = 0.87$, $P = 0.37$) as well as their interaction (all $P > 0.1$)), the lorazepam effect was only found to be significant on neutral ERPs ($F_{1, 12} = 5.00$, $P = 0.045$, $\eta p^2 = 0.294$, $(1-\beta) \approx 98.91\%$). The administration of lorazepam significantly reduced the neutral ERP amplitudes in individuals with high trait anxiety (lorazepam vs. placebo: $-0.15 \pm 0.33$ vs. $0.23 \pm 0.31\,\mu V$) (Fig. 5).

## Discussion

Extant literature has pointed to amygdala hyperreactivity at baseline as the common denominator incurring in the proclivity towards different anxiety-related illnesses. Additionally, although the association between the 5-HTTLPR and amygdala reactivity to threatening faces ever formed a cornerstone of the common view that carrying the short allele of this polymorphism is related to mechanisms of negative emotionality which confer some susceptibility towards certain affective disorders, a growing number of research yielded inconsistent results[20,21,40]. This study examined and elucidated the association between the 5-HTTLPR genotype and the propensity for trait anxiety by making use of the MMN evoked by threatening voices as the means to determine the feasibility of the tonic model of 5-HTT-dependent neural modulation. The findings revealed that S allele carriers exhibited weaker fearful MMN than noncarriers, as a function of higher baseline neutral ERPs. Thus, the weaker the fearful MMN in S allele carriers, the stronger their neutral ERPs. Fearful MMN magnitudes varied along with both fearful and neutral ERP amplitudes. Noteworthy, is that we observe that this heightened response to non-threatening voices in individuals with high trait anxiety scores can be reversed by acute anxiolytic treatment.

The S allele carriers evoked weaker eMMN than did the noncarriers, irrespective of the deviant stimuli type. Theoretically, two possible explanations can account for these findings. Firstly, one

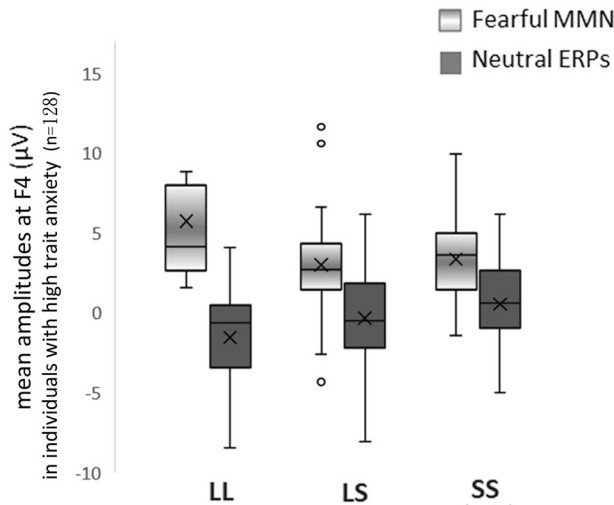

**Fig. 4 Neutral ERPs and fearful MMN in each genotype in individuals with high trait anxiety.** The S/S (neutral ERP: $0.37 \pm 0.45\,\mu V$; fearful MMN: $3.1 \pm 0.42\,\mu V$) exhibit larger neutral ERP and weaker fearful MMN than the L/S ($-0.01 \pm 0.49$; $3 \pm 0.46\,\mu V$) and L/L ($-2.23 \pm 0.99$; $4.9 \pm 0.93\,\mu V$).

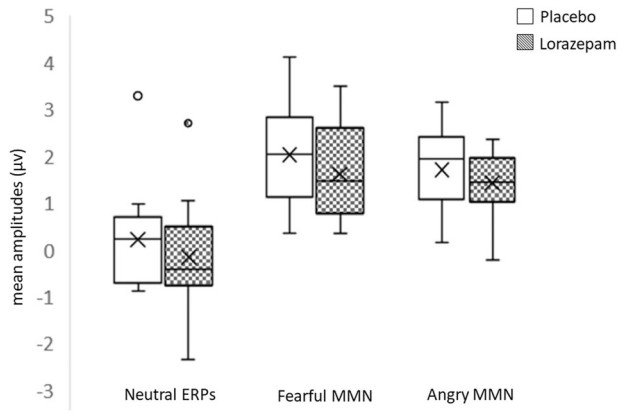

**Fig. 5 Lorazepam impacts on hyperreactivity to neutral non-threatening voices in non-clinical individuals with high trait anxiety.** While there is no significant effect of acute lorazepam treatment on neither fearful MMN ($F_{1, 12} = 1.89$, $P = 0.19$) nor angry MMN ($F_{1, 12} = 0.87$, $P = 0.37$), acute lorazepam administration significantly reduces the neutral ERP amplitudes in individuals with high trait anxiety (lorazepam vs. placebo: $-0.15 \pm 0.33$ vs. $0.23 \pm 0.31\,\mu V$; $F_{1, 12} = 5.00$, $P = 0.045$, $\eta p^2 = 0.294$, $(1-\beta) \approx 98.91\%$).

that supports the tonic model, as opposed to the phasic model, of 5-HTT-dependent modulation of neural activity[8]. While the phasic model explains the higher negative emotionality in S allele carriers as a function of higher responses to aversive stimuli *per se*, the tonic model posits a high amygdala activity is already present since baseline in these individuals, thus experiencing the unconstrained viewing of undefined stimuli as more aversive. This baseline was accessed via the amygdala modulated responses to neutral stimuli in the present study. S allele carriers exhibited decreased eMMN as a result of enhanced responses to neutral standards (see Fig. 4). When comparing the processing of neutral and negative types of emotionality, the weaker angry and fearful MMN in S allele carriers exhibited less discrepancies between them and between the neutral standards as a whole, indicating hypersensitivty in the processing of emotionally ambiguous stimuli. It is thus reasonable to assume that S allele carriers may perceive neutral voices as more aversive or threatening, consequently exhibiting a decreased ability in detecting the changes between stimuli than do noncarriers. Contrastingly, L allele carriers differentiated well the levels of negative emotionality between the fearful and angry stimuli, as reflected by their respective MMNs, and which turned to be comparable in the S/S homozygotes. Moreover, to bolster confidence in the anxiety proneness-related hyper responsiveness during the baseline condition, we attempted to replicate the findings in an independent dataset ($n = 30$, 16 males)[31]. Specifically, we identified a close relationship between eMMN to fearful vocal expressions (fearful vs. neutral voices) and left amygdala reactivity to fearful facial expressions (explicit fear vs. neutral) ($r = 0.49$, $P = 0.006$ and $r = -0.73$, $P < 0.001$, respectively) (see Fig. 3b) in a previously collected dataset[31], where the amygdala activity to neutral faces varied along with trait anxiety scores. These results converge with previous findings in depression and anxiety, and their impact on the processing of emotional facial expressions[41] and vocalizations[25], suggesting a domain-general negativity bias towards the stimuli with more ambiguous valence[42]. Secondly, MMN is a function of the N-methyl-D-aspartate receptor (NMDAR). Accordingly, pharmacological blockages of the NMDAR yield a significant reduction in MMN[29]. Previous studies attempted to observe a similar influence of serotonin on MMN through the use of serotonin receptor antagonists, and yielding findings that MMN was unaffected by these interventions, thus, concluding that the neurotransmitter had no association with MMN elicitation[43]. However, later research noted that NMDA and serotonin receptors have an inverse relationship, as serotonin is capable of blocking NMDAR stimulation, to the extent that serotonin receptor antagonists have been recently commended as possible pharmacological therapies for those affected by cognitive impairments as an effect of NMDAR deficiencies in schizophrenia, or for patients with a history of long and sustained treatment with NMDAR antagonists[44–46]. Hence, since MMN is evoked due to the activity of NMDARs[29], and in view of the 5-HTT encoding serotonin transporter protein, with S allele carriers having reduced serotonin uptake due to their less transcriptional activity[47], it seems plausible that this polymorphism will have an effect on S allele carriers in terms of evoking weaker MMN. What's more, we observed significantly reduced neutral ERP amplitudes in subjects with high trait anxiety scores after administration of the anxiolytic lorazepam (see Fig. 5). This is in line with research reporting that, when anxious states were deliberately elicited in healthy volunteers via exposure to unpredictable, aversive shocks, threat-induced anxiety prompted anxious hypervigilance, but this was reduced with subsequent administration of another benzodiazepine medication, alprazolam[28]. Past research has observed that benzodiazepine administration can reduce serotonergic activity, while increasing the inhibitory effect of the GABAergic system[48–50].

Interestingly, ethnicity may also affect the observed MMN reduction in S allele carriers. The well-known effects of the 5-HTTLPR on amygdala reactivity in Caucasian subjects might be reversed in East Asian subjects[24,51]. Similar to the distribution previously reported in East Asian populations[52], our sample of Han Chinese participants exhibited a higher proportion of the S/S genotype and a lower frequency of L/L genotype than that observed in Caucasian populations. However, and contrary to studies where the S allele was seen as conferring less susceptibility towards anxiety in Asian people, our findings demonstrated that this variation results in certain propensity towards such condition, results which are in line with studies on Caucasian populations[53,54]. In accordance with the higher scores in the STAI-T among S allele carriers[9], path analyses indicated that fearful MMN mediated the 5-HTTLPR effect on STAI-S scores while controlling the variance shared between 5-HTTLPR and STAI-S. In support of these findings, fearful MMN has been reported to be negatively correlated with social deficits in individuals with autistic traits[36]. And given that social deficits are closely coupled with anxiety[55], it is not surprising that one recent study with Han Chinese participants reported a higher level of susceptibility towards anxiety and weaker amygdala−insula functional connectivity in S/S homozygotes than in L allele carriers[56]. Thus, taking into consideration the ethnic background, we can posit that the S allele, rather than the L, is related to certain mechanisms of negative emotionality which confer some susceptibility towards anxiety-related traits in Han Chinese individuals.

Additionally, in parallel to a previous ERP study[57], we replicated the findings supporting the notion that females show stronger eMMN relative to males. This gender effect was strongest in the midline electrodes and exclusively found in subjects who were homozygous for the L allele, but not in those homozygous for the S allele or those with heterozygous alleles. Associated with this finding, previous research has observed a 5-HTTLPR × gender interaction[58]. Notwithstanding, and contrary to our results, one study in particular found this interaction to be true in females homozygous for the S allele, who were capable of recognizing negative facial expressions faster than those in the other genotype groups[59]. Nevertheless, the aforementioned studies made omission of endophenotypes in their design, and which may yield inconsistent findings[8]. Thus, we further extend the knowledge in regards to gender differences in gene-brain-behavior association studies, at the same time that we urge future researchers directly testing gene × gender interactions to include endophenotypic data in their experimental designs.

Noteworthy is that there is some discrepancy between our findings and previous works on MMN and the proclivity towards anxiety. We found that S allele carriers had both, weaker eMMN and stronger baseline neutral ERPs. A study observed that while anxious states were deliberately elicited in healthy volunteers by exposure to unpredictable, aversive shocks, threat-induced anxiety prompted anxious hypervigilance and enhanced the magnetoencephalographic counterpart of MMN (MMNm) to pure tone deviants[28]. Another research demonstrated that individuals with anxiety disorder, relative to healthy controls, showed significantly increased MMN, where MMN was elicited by complex harmonic sounds[60]. The mixed findings could be attributed to the stimuli designed to elicit MMN. eMMN involves emotional salience in addition to acoustic feature, which elicited distinct neurophysiological responses[33,61–63]. We also observed that STAI-T were positively correlated with neutral ERPs, and negatively correlated with eMMN amplitudes, but only in individuals with low anxiety scores. When interpreting these results, one has to consider the intriguing fact that—unlike for negative emotionality (fearful vs. neutral), either in ERPs being evinced by amygdala activity or in

MMN amplitudes, that showed a positive association with anxiety —for responsiveness to neutral stimuli there was also a significant positive correlation with the number of S alleles and trait anxiety scores. This shows a unique contribution of the short allele on the hypervigilance towards harmless stimuli, and in accordance with the results that negative emotionality (fearful vs. neutral) varied as a function of both neutral and negative processing (see Figs. 3, 4). The non-significant correlation between ERPs and STAI-T, as well as the results of acute anxiolytic effect on neutral ERPs in the group with high anxiety scores, could be attributed to the ceiling effect of hyper-response to non-threatening voices in this cohort scoring in the higher boundaries.

Nonetheless, our MMN findings may strengthen the cross-modal validity for the association between the *5-HTTLPR* and the propensity towards anxiety-related symptomatology. In addition to higher levels of self-reported trait anxiety, S allele carriers exhibited stronger amygdala reactivity to the passive viewing of threatening faces[17], negative pictures[19], implicit processing of negative words[15], and visuospatial matching of fearful and angry faces[41,64,65], than did noncarriers. Moreover, research examining the ERP response to a Go−NoGo task dependent on genetic variation, reported an association between the *5-HTTLPR* and inhibitory motor control[66]. In addition, the ERP response in a time window between 400 and 600 ms, associated with later semantic processing stages of happy and angry voices, was found to be reduced in S allele carriers[67]. The present neurophysiological study further demonstrates that the *5-HTTLPR* may affect threatening and non-threatening voice processing already at the pre-attentive stage.

Some limitations of this study must be acknowledged. Firstly, by using a pseudoword such as *dada*, the generalization for emotion representation might be affected. Although, studies using nonlinguistic emotional vocalizations[68] verify the passive oddball paradigm as optimal for detecting emotional salience. Secondly, unlike those studies using all of the stimuli as both standards and deviants[30], the MMN effect in this study may be potentially driven by physical stimulus characteristics. Nevertheless, we applied the same theorems as the work by Čeponienė et al.[69], as well as conducted a series of studies to test MMN for the strict task of disentangling emotional salience from physical properties[33,36,37,57,61,63,70]. Finally, future neuropharmaceutical investigations concerning gene-behavior associations are warranted, due to the small sample size in our subset utilized for the neuropharmacological intervention.

In conclusion, the present findings—which incorporate multimodal indices, including genetic, neurophysiological, biochemical, neuropharmacological, and behavioral measurements—provide evidence to corroborate the notion that the *5-HTT* has a broad impact on social cognition. Furthermore, in line with the modality-independent impact on depression and anxiety, where emotionally neutral or ambiguous stimuli are negatively biased[41,42,71], these findings suggest that the *5-HTT* affects the automatic neural processing to threatening and neutral, non-threatening voices in as early as the pre-attentive stage. What's more, a heightened baseline level of arousal can be proposed as the most likely and universal neural mechanism underlying the negative emotionality processes, which make certain individuals more susceptible to anxiety, functioning as a spectrum-like distribution from anxiety patient to include even those non-patients but who exhibit high trait anxiety scores.

## Methods

**Subjects**. This study was approved by the Ethics Committee of the National Yang-Ming University and conducted in accordance with the Declaration of Helsinki. All participants were Han Chinese. They were screened for major psychiatric illnesses (e.g., general anxiety disorder) by using the Structured Clinical Interview for DSM-IV Axis I Disorders (SCID-I), and excluded if a positive diagnosis for any of these disorders was reached, as well as due to evidence of possessing any comorbid neurological disorder (e.g., dementia, seizures), history of head injury, and/or alcohol or substance abuse or dependence within the past five years. All of them had normal bilateral peripheral hearing (pure tone average thresholds <15 dB HL) at the time of testing. A total of 188 subjects (92 males) were included in the data analysis and subdivided into three groups on the basis of genotyping results: participants possessing one copy of the S allele and one copy of the L allele were included in the L/S group, and those homozygous for the S or L allele were included in the S/S or L/L group, respectively. A written informed consent was obtained from all the participants, as well as were given monetary compensation at the end of the study.

**DNA extraction and 5-HTTLPR genotyping**. Buccal cells were harvested from the inner cheek of each subject to provide DNA for genetic testing. The DNA was extracted from buccal swabs using a QIAamp DNA Mini Kit. The procedure employed a polymerase chain reaction (PCR)-based protocol followed by restriction endonuclease digestion to identify the *5-HTTLPR* located in the promoter region of the serotonin transporter gene (*SLC6A4*) and rs25531 variants: S, LA and LG. Forward Primer: 5′-TCCTCCGCTTTGGCGCCTCTTCC-3′ and reverse primer: 5′-TggggGTTgCAggggAgATCCT-3′ (10 μM each) were used for 50 μl PCR containing about 25 ng DNA, 25 μl Taq DNA Polymerase 2× Master Mix Red (Ampliqon) and ddH2O, with an initial 5 min denaturation step at 95 ℃ followed by 35 PCR cycles of 95 ℃ (30 s), 65 ℃ (40 s), and 72 ℃ (30 s) and a final extension step of 5 min at 72 ℃. To distinguish the A/G single nucleotide polymorphism of the rs25531, we extracted 10 μl of the PCR product for digestion by FastDigest HpaII (Thermo, FD0514), an isoschizomer of MspI, a total reaction of 20 μl. These were loaded side by side on 2.5–3.0% agarose gel. For detail, agarose gel electrophoresis is conducted on the amplified PCR product and the samples after restriction endonuclease digestion. The *5-HTTLPR* amplicons length of S genotype is 469 bp, L is 512 bp. After the restriction digest the fragment lengths of alleles: SA is 469 bp, SG is 402 bp and 67 bp, LA is 512 bp, L G is 402 bp and 110 bp. Therefore, by the size difference of the PCR product, we can dissect the genotype of *5-HTTLPR*.

**Stimuli**. The auditory stimuli for the ERP recordings were emotional syllables. A young female speaker produced the spoken syllables *dada* with fearful, angry, and neutral prosodies. Within each set of emotional syllables, the speaker produced the syllables for more than ten times[37]. Emotional syllables were edited to become equally long (550-ms) and loud (min: 57 dB, max: 62 dB; mean: 59 dB) using Sound Forge 9.0 and Cool Edit Pro 2.0. Each set was rated for emotionality on a 5-point Likert-scale. Emotional syllables that were consistently identified as the extremely fearful and angry, as well as the most emotionless were selected as the fearful, angry and neutral stimuli, respectively. The ratings on the Likert-scale (mean ± SD) for the fearful, angry, and neutral syllables were 4.34 ± 0.65, 4.26 ± 0.85, and 2.47 ± 0.87, respectively (see for validation[31–33,36–38,57,61–63,70]).

**Procedures**. After recording ERPs, the State-Trait Anxiety inventory (STAI) was administered to the participants as to determine their self-reported anxiety levels[72]. State anxiety (STAI-S) indicates anxiety in specific situations, and trait anxiety (STAI-T) determines anxiety as a general trait. Given that scoring in the top range of the STAI-T suggests these participants might be experiencing some type of undiagnosed or previously unreported anxiety disorder, we used a structured clinical interview to ensure that none of the subjects had any evidence of such conditions.

**EEG apparatus and recordings**. The ERP recordings were conducted in an electrically shielded room. Stimuli were presented binaurally via two loudspeakers placed on the right and the left side of the subject's head. The sound pressure level (SPL) peaks of different types of stimuli were equalized to eliminate the effect of the angry stimuli's substantially greater energy. The mean background noise level was around 35 dB SPL. During recording, participants were required to watch a muted movie with subtitles, while task-irrelevant emotional syllables in oddball sequences were presented, as to control for attentional modulation. Participants were told to ignore the task-irrelevant emotional syllables. The passive oddball paradigm employed the fearful and angry syllables as deviants, and the neutral syllables as standards. There were two blocks. Each block consisted of 450 trials, of which 80% were neutral syllables, 10% were fearful syllables, and the other 10% were angry syllables. The sequences of stimuli were quasi-randomized such that successive deviant stimuli were avoided. The stimulus-onset-asynchrony was 1200 ms, including a stimulus length of 550 ms and a 650 ms interstimulus interval.

The electroencephalogram was continuously recorded from 32 scalp sites using electrodes mounted in an elastic cap, and positioned according to the modified International 10−20 system, with the addition of two mastoid electrodes. The electrode at the right mastoid (A2) was used as the on-line reference. Eye blinks and eye movements were monitored with electrodes located above and below the left eye. The horizontal electro-oculogram was recorded from electrodes placed 1.5 cm lateral to the left and right external canthi. A ground electrode was placed on the forehead. Electrode/skin impedance was kept <5 kΩ. Channels were

re-referenced off-line to the average of left and right mastoid recordings [(A1 + A2)/2]. Signals were sampled at 500 Hz, band-pass filtered (0.1–100 Hz), and epoched over an analysis time of 900 ms, which included 100 ms of pre-stimulus used for baseline correction. An automatic artifact rejection system excluded from the average all trials containing transients exceeding ±70 μV at recording electrodes and exceeding ±100 μV at the horizontal EOG channels. Furthermore, the quality of ERP traces was ensured by careful visual inspection in every subject and trial, and by applying an appropriate digital, zero-phase shift band-pass filter (0.1–50 Hz, 24 dB/octave). The first ten trials were omitted from the averaging in order to exclude unexpected large responses elicited by the initiation of the sequences. The paradigm was edited using the MatLab software (The MathWorks, Inc., USA). Each event in the paradigm was associated with a digital code that was sent to the continuous EEG, allowing off-line segmentation and average of selected EEG periods for analysis. The ERPs were processed and analyzed using Neuroscan 4.3 (Compumedics Ltd., Australia).

**Statistical analyses**. The MMN amplitudes were defined as the average within a 50 ms time window surrounding the peak at the electrode sites F3, Fz, F4, C3, Cz, and C4. The peak was defined as the largest negativity of the difference between the deviant and standard ERPs during a period of 150–350 ms after stimulus onset. Only the standards before the deviants were included in the analysis. MMN was statistically analyzed using a mixed ANOVA comprising gender (male or female) and genotype (L/L, L/S, and S/S) as the between-subjects factor, and the deviant type (fearful or angry), coronal site (left, midline, and right) and anterior–posterior site (frontal or central) as the within-subjects factors. Degrees of freedom were corrected using the Greenhouse−Geisser method. A post hoc comparison was performed only when preceded by significant main effects. Statistical power $(1−β)$ was estimated by G*Power 3.1 software[73]. Path analyses with structural equation modeling (SEM) were performed to examine the relationships and directionality among 5-HTTLPR, eMMN, and anxiety using the Bayesian information criterion (BIC)[74], which entailed quantifying model evidence (favoring fit accuracy and penalizing complexity). Statistical analyses were performed using SPSS 17.0 and IBM SPSS AMOS 23.0 (see Supplementary materials for further details).

**Follow-up neuropharmacological examination**. Thirteen volunteers with high trait anxiety were willing to participate in the follow-up neuropharmacological examination. A double-blind, crossover, within-subjects design was employed. In one session, participants received a single 0.5 mg dose of anxiolytic (lorazepam tablets 0.5 mg, aka ATIVAN) 2 h before the EEG experiment, and, in the other session, they received a single dose of placebo also 2 h before the EEG experiment. There was at least one-week interval between both sessions. The sequence of placebo and lorazepam administration was counter-balanced between subjects: half of the participants went first through the placebo session, and the other half went first through the lorazepam session. The participants underwent the same EEG recording and eMMN protocol as those previously mentioned. To minimize the effect of circadian rhythm and maximize the test–retest reliability[32], both sessions were recorded in the mid-afternoon (around 15:30 PM). The STAI and Beck Anxiety Inventory (BAI) were administered to these subjects to determine their self-reported anxiety levels 1.5 h after receiving either the placebo or lorazepam, and right before the EEG recording.

**Statistics and reproducibility**. To recapitulate, DNA data was extracted from buccal swabs using a QIAamp DNA Mini Kit. The procedure employed a polymerase chain reaction (PCR)-based protocol followed by restriction endonuclease digestion to identify the 5-HTTLPR located in the promoter region of the serotonin transporter gene (SLC6A4) and rs25531 variants. For EEG data, the MMN amplitudes were defined as the average within a 50-ms time window surrounding the peak at the electrode sites F3, Fz, F4, C3, Cz, and C4. The peak was defined as the largest negativity of the difference between the deviant and standard ERPs during a period of 150–350 ms after stimulus onset. Only the standards before the deviants were included in the analysis. MMN was statistically analyzed using a mixed ANOVA comprising gender (male or female) and genotype (L/L, L/S, and S/S) as the between-subjects factor, and the deviant type (fearful or angry), coronal site (left, midline, and right) and anterior–posterior site (frontal or central) as the within-subjects factors. Degrees of freedom were corrected using the Greenhouse−Geisser method. A post hoc comparison was performed only when preceded by significant main effects. Statistical power $(1−β)$ was estimated by G*Power 3.1 software. Path analyses with structural equation modeling (SEM) were performed to examine the relationships and directionality among 5-HTTLPR, eMMN, and anxiety using the Bayesian information criterion (BIC), which entailed quantifying model evidence (favoring fit accuracy and penalizing complexity). Statistical analyses were performed using SPSS 17.0 and IBM SPSS AMOS 23.0 (see supplementary materials for further details).

**Reporting summary**. Further information on research design is available in the Nature Research Reporting Summary linked to this article.

## Data availability

All data needed to evaluate the conclusions in the present paper are included in the paper and/or the Supplementary Materials via the open source repository fig share (https://doi.org/10.6084/m9.figshare.11815875). Additional data related to this paper is available from the corresponding authors upon reasonable request.

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

## Acknowledgements

We thank Chenyi Su and Yu Huang for assisting with the data collection. The study was funded by the Ministry of Science and Technology (MOST 108-2410-H-010-005-MY3; 108-2636-H-038-001-; 109-2636-H-038-001-; 108-2636-B-038-001-; 109-2636-B-038-001-), National Yang-Ming University Hospital (RD2019-003; RD2020-003), Taipei Medical University (DP2-108-21121-01-N-03-03; TMU108-AE1-B25), and the Brain Research Center, National Yang-Ming University from The Featured Areas Research Center Program within the framework of the Higher Education Sprout Project by the Ministry of Education (MOE) in Taiwan (108BRC-B501).

## Author contributions

C.C. and Y.C. conceived and conceptualized the study. R.M.M., C.C., T.T.L., C.Y.C., and C.Y.Y. performed the experimental work in order to acquire data. All authors performed different types of analyses required for the present study (from EEG, fMRI, and genetic analyses, to different statistical methods). R.M.M., C.C., and Y.C. conducted the necessary literature reviews and drafted the first manuscript. All authors contributed towards the revision and writing of the final draft.

## Competing interests

The authors declare no competing interests.
