## [Peer Review File · Communications Biology]

Reviewers' comments:

Reviewer #1 (Remarks to the Author):

In this study, the authors conduct an explicit test of the 5-HTTLPR tonic versus phasic model of activation in a cohort of non-patients, using the mismatch negativity (MMN) paradigm, re-analysis of fMRI data from an independent samples, and pharmacological intervention using application of the anxiolytic lorazepam in a double-blind, crossover, within-subjects study design. The results are consistent with the tonic model of 5-HTTLPR activation showing elevated reactivity to neutral (and reduced activation to emotional) stimuli, which can be normalized by lorazepam, and can be localized to the amygdala.

The study has a number of strengths. First, it is tightly constrained by theoretical framework that juxtaposes two distinct predictions about the relationship of 5-HTTLPR genotype and emotional stimulus reactivity. The presentation of the theory was well-balanced in the introduction, which included both supporting and non-replicating studies. An explicit test of the model is of very high interest to a broad readership of emotion and genetic researchers, because the role of 5-HTTLPR has been highly controversial, and few theoretical models of its function have been proposed, and rarely tested. Second, the EEG part of the study is well-powered with a sample size of hundred and eighty-eight individuals. Genotyping of this sample was conducted in a manner that included a tri-allelic analysis of the A/G SNP located within the long allele. Third, the authors extend their findings to amygdala imaging, by conducting a re-analysis of previously published data. For the convenience of reading, it would be helpful to include the sample size of this independent sample, so that readers do not need to retrieve this information from the earlier publication. Fourth, the inclusion of pharmacological intervention data adds an experimental manipulation that is rarely seen in gene-behavior association studies. One concern is that the sample size is very small ($N = 13$), but this concern is alleviated by the double-blind within-subjects study design.

Reviewer #2 (Remarks to the Author):

The study reports that 5-HTTLPR short-allele carriers had significantly increased baseline ERPs and reduced fearful MMN, phenomena which can nevertheless be reversed by acute anxiolytic treatment. Such evidence is in principle novel and currently interesting on the extant literature. The overall design, implementation and interpretation of the findings is accurate, but the overall theoretical approach which perceives 5-HTTLPR as 'per se' risk factor is rather dated and biased towards the biomedical model of neuropsychiatric disease without acknowledging the psycho-social contributions on the behaviours of interest. More specifically, the below areas are highlighted as problematic, but there are significant changes that need to be made throughout the paper to reflect an overall shift of the model in which the study is based:

(lines 26-74) Authors refer to 5-HTT as 'risk factor' in line with a considerable amount of the relevant literature that suggests per se risk factors. Currently, such accounts are not supported from current models of psychopathology (e.g., see Differential Susceptibility Hypothesis), or meta-analytical studies on 5-HTTLPR alone. Therefore the authors are requested to 'temper' the overall assumed links between 5-HTTLPR and increased risk for psychopathology, and refer instead to the mechanism as 'susceptibility' or 'sensitivity' factor, accounting the same time for current/modern views on psychopathology.

(lines 93-96). The authors only briefly and loosely referred to the processing chain from the ventral auditory pathway to 'brain structures' involved in emotion and cognition. The relevant literature is

central and critical to the hypothesis of the study, therefore the authors are encouraged to expand considerably around the underlying neurobiological pathways of the area of inquiry.

Reviewer Recommendation: Resubmission

Antonios I. Christou, PhD, SFHEA, AFBPsS

RESPONSES TO THE REVIEWERS

Ms. # COMMSBIO-19-1532-TR1

We are very thankful to the Editor and the Reviewers for taking their precious time to provide a number of constructive comments. The manuscript has been substantially revised per the Editor and Reviewers' suggestions.

Reviewer #1:

In this study, the authors conduct an explicit test of the 5-HTTLPR tonic versus phasic model of activation in a cohort of non-patients, using the mismatch negativity (MMN) paradigm, re-analysis of fMRI data from an independent samples, and pharmacological intervention using application of the anxiolytic lorazepam in a double-blind, crossover, within-subjects study design. The results are consistent with the tonic model of 5-HTTLPR activation showing elevated reactivity to neutral (and reduced activation to emotional) stimuli, which can be normalized by lorazepam, and can be localized to the amygdala.

The study has a number of strengths. First, it is tightly constrained by theoretical framework that juxtaposes two distinct predictions about the relationship of 5-HTTLPR genotype and emotional stimulus reactivity. The presentation of the theory was well-balanced in the introduction, which included both supporting and non-replicating studies. An explicit test of the model is of very high interest to a broad readership of emotion and genetic researchers, because the role of 5-HTTLPR has been highly controversial, and few theoretical models of its function have been proposed, and rarely tested. Second, the EEG part of the study is well-powered with a sample size of hundred and eighty-eight individuals. Genotyping of this sample was conducted in a manner that included a tri-allelic analysis of the A/G SNP located within the long allele. Third, the authors extend their findings to amygdala imaging, by conducting a re-analysis of previously published data. For the convenience of reading, it would be helpful to include the sample size of this independent sample, so that readers do not need to retrieve this information from the earlier publication. Fourth, the inclusion of pharmacological intervention data adds an experimental manipulation that is rarely seen in gene-behavior association studies. One concern is that the sample size is very small ($N = 13$), but this concern is alleviated by the double-blind within-subjects study design.

- Many thanks for your precious time. We included the sample size for the independent sample where needed: “($n = 30$, 16 males)” (p. 11, 16, 44)

Reviewer #2:

The study reports that 5-HTTLPR short-allele carriers had significantly increased baseline ERPs and reduced fearful MMN, phenomena which can nevertheless be reversed by acute anxiolytic treatment. Such evidence is in principle novel and currently interesting on the extant literature. The overall design, implementation and interpretation of the findings is accurate, but the overall theoretical approach which perceives 5-HTTLPR as 'per se' risk factor is rather dated and biased towards the biomedical model of neuropsychiatric disease without acknowledging the psycho-social contributions on the behaviours of interest. More specifically, the below areas are highlighted as problematic, but there are significant changes that need to be made throughout the paper to reflect an overall swift of the model in which the study is based: (lines 26-74) Authors refer to 5-HTT as 'risk factor' in line with a considerable amount of the relevant literature that suggests per se risk factors. Currently, such accounts are not supported from current models of psychopathology (e.g., see Differential Susceptibility Hypothesis), or meta-analytical studies on 5-HTTLPR alone. Therefore the authors are requested to 'temper' the overall assumed links between 5-HTTLPR and increased risk for psychopathology, and refer instead to the mechanism as 'susceptibility' or 'sensitivity' factor, accounting the same time for current/modern views on psychopathology.

- Many thanks for your precious time and constructive criticism. Besides adding the lines indicated below, we also rephrased and reframed the phrases where needed along the whole manuscript.
- “This assumption arises from the Differential Susceptibility Hypothesis, which explains how individuals experience life circumstances and events in a differing manner, dependent on pre-existing biological factors which may result in certain predispositions.” (p. 3)
- “But given the impact of “epigenetic” mechanisms that encode environmental information from both internal and external bodily sources, this Gene-Brain interaction may render various degree of sensitivity towards threat processing” (p. 4-5)

(lines 93-96). The authors only briefly and loosely referred to the processing chain from the ventral auditory pathway to 'brain structures' involved in emotion and cognition. The relevant literature is central and critical to the hypothesis of the study, therefore the

authors are encouraged to expand considerably around the underlying neurobiological pathways of the area of inquiry.

- Many thanks for pointing this out. We added the pertinent information where needed. We added the pertinent information where needed: “[...] a processing chain that proceeds from the primary auditory pathway to brain structures implicated in cognition and emotion –e.g., the orbitofrontal cortex, amygdala, superior temporal gyrus and sulcus–, as well as in the saliency network (insula), has been revealed.” (p. 6).

REVIEWERS' COMMENTS:

Reviewer #2 (Remarks to the Author):

The authors have made considerable changes on the revised manuscript as they have been instructed by both the reviewers.